Engaging inexpensive hands-on activities using Chlamydomonas reinhardtii (a green micro-alga) beads to teach the interplay of photosynthesis and cellular respiration to K4–K16 Biology students

Mitra Mautusi mmitra@westga.edu 1
Broom Sara Michelle 1
Pinto Kysis 1
Wellons Sovi-Mya Doan 2 3
Roberts Ariel Dominique 1
1 Biology Department, University of West Georgia , Carrollton , GA , USA
2 The Heritage School , Newnan , GA , USA
3 Department of Biological Sciences, Dartmouth College , Hanover , NH , USA
Sotelo-Mundo Rogerio
Electronic publication date: 2020 Aug 25
Publication date: 2020
Volume: 8
Electronic Location ID: e9817
Received 2020 Jun 3; Accepted 2020 Aug 5
Copyright: ©2020 Mitra et al.
Copyright year: 2020
Copyright holder: Mitra et al.
License: This is an open access article distributed under the terms of the Creative Commons Attribution License, which permits unrestricted use, distribution, reproduction and adaptation in any medium and for any purpose provided that it is properly attributed. For attribution, the original author(s), title, publication source (PeerJ) and either DOI or URL of the article must be cited.
License URL: https://creativecommons.org/licenses/by/4.0/

Keywords: Chlamydomonas, Photosynthesis, Cellualr respiration, Green micro-alga, Algae bead bracelets, Algal beads, Light-powered algae bracelets, Plant biology, Biology education

Funding: American Society of Plant Biologists’ Plant-BLOOME grant University of West Georgia’s College of Science and Mathematics This work was supported by the American Society of Plant Biologists’ Plant-BLOOME grant (2018) and by the 2018 University of West Georgia’s College of Science and Mathematics’ faculty research grant. The funders had no role in study design, data collection and analysis, decision to publish, or preparation of the manuscript.

==============================
Background

Photosynthesis and cellular respiration play major roles in energy metabolism and are important Life Science topics for K16 Biology students. Algae beads are used for photosynthesis and cellular respiration labs. Currently there are a few companies that sell biology educational kits for making algae beads using non-motile green micro-algae to introduce students to photosynthesis. These kits are expensive and, do not come with detailed guidelines for trouble shooting and customizations for different grade levels. Chlamydomonas reinhardtii is a motile green micro-alga and is an excellent model system for photosynthesis studies. In this article, we are presenting the work conducted in the student-driven, American Society of Plant Biologists-funded, Plant-BLOOME educational outreach project. This project is a supervised collaborative effort of three undergraduates and one high school student. We have generated a protocol which can be used to make Chlamydomonas beads. We have used these beads to design two simple and inexpensive plant biology hands-on activities. These laboratory activities have been customized to teach the interplay of photosynthesis and cellular respiration to K4–K16 Biology students.

Methods

Chlamydomonas beads were used for two different laboratory activities that involved monitoring pH changes over time using a pH indicator. Our first activity centers on making and, using light-powered algae bead bracelets to monitor dramatic color/pH changes over time when exposed to darkness or light. Our second activity employs strain-specific algae beads with approximately equal cell numbers to conduct comparative photosynthesis and cellular respiration studies in two Chlamydomonas strains namely, wild type, 4A+ and, a high light-sensitive, photosynthetic mutant, 10E35/lsr1a.

Results

We optimized our experimental protocol using algae beads in a 5.5 mL screw capped glass vials before performing the same experiment in algae bead bracelets. We found that the algal cell density/bead, water type used in the experiment and, the duration of dark exposure of algal beads can affect successful implementation of the lab activities. Light-powered algae bead bracelets showed dramatic color/pH changes within 3 h upon exposure to light or darkness. These bracelets could be switched back and forth between darkness and light multiple times within 48–72 h to display color/pH changes, provided prior dark exposure time did not exceed 9 h. Our comparative studies of photosynthesis and cellular respiration in 10E35 and in 4A+ showed that relative respiration rate and photosynthetic rate is higher and lower in 10E35, respectively, compared to that in 4A+. Additionally, 10E35 failed to display the expected photosynthesis-induced pH/color changes in the light after prolonged exposure to darkness which indicated that prolonged dark exposure of 10E35, hindered photosynthesis.

Introduction

Photosynthesis, an energy requiring anabolic process, comprises of two sets of reactions that occur in the chloroplast: Light reaction and Calvin cycle. In the light reaction, solar energy captured by photosynthetic pigments is used to photolyze water into electrons and protons. These protons and electrons are ultimately used to generate ATP (adenosine triphosphate), a reducing power and oxygen in the light reaction (Tymoczko, Berg & Stryer, 2015). ATP and the reducing power generated in the light reaction and water, are used in the Calvin cycle to reduce atmospheric carbon dioxide to sugar (Tymoczko, Berg & Stryer, 2015). Aerobic cellular respiration is a catabolic energy releasing process that oxidizes fixed carbon to generate ATP (Tymoczko, Berg & Stryer, 2015). Oxygenic photosynthesis provides not only fixed carbon that is utilized by cellular respiration for energy production but, it is also the only source for generation of oxygen on a mass scale on Earth to support life (Tymoczko, Berg & Stryer, 2015). Hence, every Biology students should have a broad understanding of these two complementary, life-supporting, fundamental biochemical reactions. These two biochemical reactions are listed in the Next Generation Science Standards (NGSS) Life Science core idea LS1C: From Molecules to Organisms: Structures & Processes. LS1C aligns with principles 1, 2, 3, 5, 10 and 11 of the 12 Principles of Plant Biology listed by the American Society of Plant Biologists (American Society of Plant Biologists, 2020; Article S1).

Guidelines for photosynthesis and cellular respiration laboratories using non-motile green micro-algae beads are available on the websites of Carolina Biological (Burlington, NC), Bio-Rad (Hercules, CA) and Gene Technology Access Center (GTAC; Victoria, Australia) for classroom use. (Carolina Quicktips Making Algae Beads, Carolina Biological; Bio-Rad, 2020; Algae Immobilised in Alginate balls, GTAC, 2016). Traditionally, non-motile algae like Chlorella, Ankistrodesmus and Scenedesmus have been used to generate algae beads as non-motile algae can be trapped and immobilized easily (Carolina Quicktips Making Algae Beads, Carolina Biological; Algae Immobilised in Alginate balls, GTAC, 2016; Bio-Rad, 2020). Chlamydomonas reinhardtii is motile green micro-alga and is an excellent model system for photosynthesis and bioenergy research (Merchant et al., 2007; Scranton et al., 2015; Radakovits et al., 2010). At our research laboratory we use Chlamydomonas as an experimental system to study photo-protection and photosynthetic pigment metabolism. Our ASPB-funded Plant-BLOOME educational outreach student-driven project centers on designing new educational hands-on activities using exclusively, Chlamydomonas and not any other alga.

There is one recent report of immobilization of a Chlamydomonas strain for photobiohydrogen production under anaerobic sulfur-deprived conditions in different types of photobioreactors (Canbay, Kose & Oncel, 2018). Low yield of biofuel from large-scale algal cultures in bioreactors is a major problem. Sustainable bioenergy production can be improved by immobilizing motile algae like Chlamydomonas and Botryococcus sp. that are employed for bioenergy research (Canbay, Kose & Oncel, 2018; Radakovits et al., 2010).

Our educational trips to schools in Georgia and our participation at the education booths at the NSTA meeting in Atlanta in 2018 and at the Plant Biology 2019 in San Jose, CA, clearly showed us that K6- K16 students (and even educators) love to make algae beads (Text S1). Bead-making activity is excellent for student engagement in classrooms, which cannot be achieved by using commercially purchased pre-made beads. Unfortunately, anonymous teacher and student surveys that we collected cannot be shared with public because our institution did not submit IRB application materials for this project. Glimpses of our educational outreach activities can be found at several available links shown in Text S1.

Commercial kits from Bio-Rad or other vendors are costly when one considers how many students can be served per commercial kit and the duration of the time the kit can be used in classrooms (see Materials and Methods and Text S2 for detailed cost calculation). Commercial educational kits often do not work well, uses beads with short shelf lives, takes long time to show color change and, sometimes comes with erroneous instructions (For example: Carolina instruction sheet instructs educators to grow dense cultures of Chlorella for 3–4 weeks before harvesting cells for bead-making. This means the company is instructing educators to make algal beads using a culture that is in the late stationary phase; https://www.carolina.com/pdf/activities-articles/carolina-qt-making-algae-beads-cb814921806.pdf). Technical resources that comes with these kits lack specific guidelines for optimizing the experiment and troubleshooting. Hence a well–defined protocol with proper detailed guidelines for conducting lab activities & managing class times and, information for acquiring lab materials inexpensively, will be useful Biology educators at schools and institutions that have very limited resources and funding.

Calcium alginate is used to trap and immobilize living cells in industrial procedures (Carolina Quicktips Making Algae Beads, Carolina Biological; Algae Immobilised in Alginate balls, GTAC, 2016; Bio-Rad, 2020). For example, immobilized non-motile colonial algae are being tested for biofuel production, immobilized yeast cells are being used for alcoholic wine fermentation, and immobilized bacterial cells are being used for water disinfection (Kröger & Müller-Langer, 2012; Gotovtsev et al., 2015). To entrap algae in beads, cell suspension and 2% sodium alginate are mixed at a specific ratio and added drop-wise to chilled calcium chloride solution. Calcium ions link the alginate monomers together to make a gel-like polymer of calcium alginate which trap cells and immobilize them in beads. These algal beads can be used for biological experiments or other biotechnological applications.

Cellular respiration oxidizes organic chemicals and releases CO2 into the environment irrespective of presence/absence of light and, photosynthesis converts CO2 into fixed carbon only in the presence of light (Tymoczko, Berg & Stryer, 2015). Cellular respiration in live cells in the beads will release CO2 that will dissolve in water in which the beads are immersed to generate carbonic acid (Bio-Rad, 2020; Algae Immobilised in Alginate balls, GTAC, 2016). Conversely in the light, photosynthesis in the algal cells in the beads will remove CO2 from the water surrounding the beads (Bio-Rad, 2020; Algae Immobilised in Alginate balls, GTAC, 2016). In the light, cellular respiration is still going on in the cells, but the net use of CO2 by photosynthesis vastly outweighs the CO2 released during cellular respiration, unlike that in the dark. Hence pH of the water will be acidic in the dark and alkaline in the light. In the two activities designed by us, students will monitor photosynthesis and cellular respiration-induced pH changes in the water by color changes of a pH indicator as well as by measuring the pH with pH testing strips and/or a pH electrode.

We have generated a detailed protocol of making Chlamydomonas beads and two simple plant biology hands-on activities. These laboratory activities were used to teach the interplay of photosynthesis and cellular respiration to Biology students in nine schools and two universities in Georgia in a fun and engaging way. The presented educational work is a product of supervised collaborative efforts of three undergraduate students and one high school student in Georgia, USA. In the two designed laboratory activities students make Chlamydomonas beads and use these beads to conduct their own independent experiments. In the first lab activity students make light-powered green algae bead bracelets and use these algae bead bracelets to perform time course experiments in light and dark to study the interplay of photosynthesis and cellular respiration. In the second activity, students compare relative ratios of photosynthesis and cellular respiration in a Chlamydomonas wild type (4A+) and a chlorophyll-deficient, high-light sensitive mutant strain, 10E35/lsr1a, using strain-specific algae beads. 10E35 is a random insertional mutant generated by our research lab with a mutation in a novel functionally uncharacterized gene, LSR1 and is the center of an on-going research project at our laboratory (Nguyen et al., 2017; Article S2).

We present in this article our protocol for making Chlamydomonas beads (including some preliminary testing data that helped us to refine the protocol), two new plant biology teaching tools and sample teaching resources for educators. We hope that the teaching resources will help plant biology educators to customize the labs according to grade level, availability of resources, and allow better time management in classrooms. The designed lab activities support active learning and contributes toward the following: (1) NGSS Science and Engineering Practice: Developing and using models; Planning and carrying out investigations and, (2) NGSS Core Idea: Life Science LS1C: From molecules to organisms: Structures and Processes.

Materials & Methods

Material information for educators

Information (vendors and catalog numbers) for ordering specific items related to the project like algal strains, algal growth media, inoculating loops, flasks, plastic transfer pipettes, Eppendorf tubes, pH indicators, pH test strips, bracelet tubing, glass vials, yarns for bracelet braids, sodium alginate, calcium chloride, counting chambers etc. are given on pages 1–6 in Text S2. On pages 5–6 in Text S2, we have shown the pricing of the basic items that one will need to start the lab and the cost comparison of our protocol Vs. the Bio-Rad Photosynthesis and Cellular respiration kit for general Biology. The cost comparison shows that our protocol is inexpensive and will serve more students over a longer period than the Bio-Rad kit (Text S2).

Algal media and cultures

Chlamydomonas wild type strain 4A+ (CC- 4051 4A+ mt+) strain was a gift from Dr. Krishna K. Niyogi (UC Berkeley, CA). 10E35/lsr1a (light-sensitive related 1a) is a random insertional nuclear mutant generated by our lab which has a mutation in a novel gene, LSR1 encoding a protein of unknown function (Nguyen et al., 2017; Article S2). 4A+ and 10E35 strains were maintained in the lab on Tris-Acetate Phosphate (TAP) agar media plates (Text S2) in dim light intensities (15–20 µmol m−2s−1) at 25 °C. A starter culture of 4A+ was started approximately 11–12 days ahead of the lab activity by inoculating 10 mL of liquid TAP media in a 50 mL flask with 4A+ cells from a 5-day old TAP agar media plate (Text S2). After 5 days of growth, 1 mL of the starter culture was used to inoculate 300 mL of fresh TAP media in a 1L flask. The TAP liquid 4A+ culture was grown for 6–7 days for dense dark green growth. 10E35 grows slower than 4A+. Hence 10E35 liquid TAP cultures should be started at least 3–4 days before starting the 4A+ liquid TAP cultures. Algal liquid cultures were grown under 25 °C under continuous illumination of 80–100 µmol photons m−2 s−1 provided by the combined light intensities of four to six cool white fluorescent lights. Cultures were shaken continuously on an open-air orbital shaker at a speed of 150–180 rpm to ensure a uniform illumination of the cells and to prevent cells from settling down. Light intensities were measured using a LI-250A Light Meter (LI-COR, Inc., Lincoln, NE).

Preparation of 2% sodium alginate and 3% calcium chloride solutions

2 g of sodium alginate (Fisher Scientific, Waltham, MA) was dissolved in 100 mL of E-pure water overnight at room temperature by stirring at a speed of 400 rpm using a magnetic stirrer. [Note: sodium alginate forms a very viscous solution when dissolved at 1.5%–4%]. 2% sodium alginate solution was stored at room temperature. 30 grams of calcium chloride was dissolved in 1,000 mL of E-pure water and stored at 4 °C in a fridge.

Cell counts

Cell density (number of cells per mL of the culture) was determined before harvesting Chlamydomonas cells from the TAP liquid culture to estimate the volume of culture needed to harvest specific number of cells per 50 mL falcon tube. Cell density was calculated by counting the cells using a Hausser Scientific Bright-Line™ Counting Chamber (Hausser Scientific, Philadelphia, PA). A basic protocol on how to use a hemocytometer in a classroom setting is available at https://www.ruf.rice.edu/∼bioslabs/methods/microscopy/cellcounting.html. It is to be noted that cell counting is optional. School teachers who do not have access to a hemocytometer/counting chamber, can grow algae culture for 6–7 days and then harvest the cells to make beads. Additionally, teachers can match the green color of the beads with that shown in our article figures.

Preparation of Chlamydomonas 4A+ and 10E35 beads

A detailed version of the Chlamydomonas bead-making protocol (including trouble shooting) is available at https://www.protocols.io/view/making-inexpensive-light-powered-chlamydomonas-rei-bgpyjvpw. Chlamydomonas strain 4A+ or 10E35 cells were harvested by spinning down dense TAP liquid strain-specific cultures at 1,000–1,500 g for 3 min in a benchtop centrifuge. The supernatant was discarded and the cell pellet was collected. Harvesting 100 mL of dense Chlamydomonas culture generated 200–300 beads of 4–5 mm in diameter. 2% well mixed-sodium alginate was added to the cell pellet in a 4:1 or 5:1 ratio (depending on the total number of cells harvested; see results and detailed protocol on https://www.protocols.io/view/making-inexpensive-light-powered-chlamydomonas-rei-bgpyjvpw). The algae and 2% sodium alginate were gently mixed till the entire cell pellet was completely resuspended without any visible cell clumps. Maximum number of total cells used for resuspension in sodium alginate was either 395 × 106 or 790 × 106 cells depending on the experiment (see result section). We resuspended the cell pellets containing 395 × 106 cells and 790 × 106 cells in 5 mL of sodium alginate to get an approximate final cell density of 66 × 106 cells/mL and 132 × 106 cells/mL in the cell suspension, respectively. 1 mL of sodium alginate-algal cell suspension gave us approximately 32–35 beads depending on pipetting techniques. Hence the cell suspension with cell density of 66 × 106 cells/mL will form beads that have approximately 1.89 × 106–2 × 106 cells/bead while the cell suspension with cell density of 132 × 106 cells/mL will have 3.77 × 106– 4.1 × 106 cells/bead. We used 8 beads of similar sizes (4–5 mm in diameter) for glass vial experiments.

The algae-sodium alginate mix was added drop wise steadily and quickly with uniform pipetting by using a micropipette or a plastic transfer pipette into a beaker of pre-chilled 3% calcium chloride kept on ice. If pipetting is not smooth and regular and, the algae-sodium alginate mixture is not mixed by swirling in between pipetting, irregular shaped and beads with different cell numbers/bead (light green and dark green beads) will form (Fig. S1). As soon as the algae-sodium alginate mixture touched the chilled calcium chloride liquid surface, the mixture solidified into tiny beads. The calcium chloride beaker containing the beads were kept on ice for 10–15 min to allow complete solidification of the algal beads.

The beads are separated from the calcium chloride solution by filtering through an oil strainer. Algal beads on the strainer were washed with tap water. The beads were kept temporarily in a petri dish containing small amount of tap water till the bracelets were made. Surplus beads were stored in tap water in a beaker for future use within 1–2 days. Algae bead making demonstration video clips are available at: https://youtu.be/u4BbZ29qlWQ and at https://youtu.be/eIxbzeHW8IM.

Preparation of Chlamydomonas 4A+ bead bracelet

Flexible tubing was cut into 10 pieces, each 5 inches long. Caps of 1.5 mL Eppendorf tubes were cut off with a scissor. De-capped Eppendorf tube was used to plug the ends of the bracelet tubing (one de-capped tube at each end of the cut tubing). Colorful cotton yarn was cut according to the wrist width, intertwined and yarn braids were made. One braid was looped tightly onto the mouth of each de-capped Eppendorf tube at each end of the bracelet. Next, one end of the bracelet tubing was unplugged by removing the de-capped Eppendorf tube that was sealing the end. About 3.5 mL of tap water [pH 7.2–7.3] was introduced into the bracelet flexible tubing. 15–38 algae beads (depending on the experiment) were gently introduced into the water inside the tubing. 8–10 drops of the bicarbonate indicator (Carolina Biological, Burlington, NC) were added into the water in the tubing and the end of the tubing was plugged back with the de-capped Eppendorf tube. Precautions were taken to avoid acidic or alkaline contamination of the flexible tubing, plastic spoon, transfer pipettes, petri dishes etc. used in our experiments, since the bicarbonate indicator is not directly specific to gases like carbon dioxide. About 0.5 cm–1 cm air gap was left at each end inside the tubing to provide enough air for cells. The bracelet was imaged and the pH of the water inside the bracelet was measured using pH testing strips (Fisher Scientific, Waltham, MA) before shifting it to light or to darkness for the lab activity. Individual experiments described below were performed with the same batch of beads. A detailed version of the protocol is available at https://www.protocols.io/view/making-inexpensive-light-powered-chlamydomonas-rei-bgpyjvpw. Demonstration of algae bead bracelet making video clips available at: https://youtu.be/A7VIjLDGSCc and https://youtu.be/vh_1ASpQgS8 and https://youtu.be/enctr0yhWQ8.

Light and dark exposure experiments with Chlamydomonas bead bracelets

For the constant light/dark exposure experiment, one bracelet was kept under 150–200 µmole m−2s−1 light intensity [equivalent to the combined light intensities of 12 to 14 cool white fluorescent lights] and another one was kept in the dark inside a lab cabinet drawer. After 3 h of light/dark exposure, bracelets were imaged. pH of the water inside the bracelets were measured using pH testing strips (Fisher Scientific, Waltham, MA).

For dark shift experiment, the bracelet was first light-adapted for 4 h and then shifted to darkness. For light shift experiment, the bracelet was dark-adapted for 4 h and then shifted to light. After every 1 h over a period of 4 h during light exposure or over a period of 3 h during dark exposure, the bracelet was imaged to monitor the carbon dioxide percentage change inside the bracelet tubing. The carbon dioxide percentage change is monitored indirectly by the color changes of the bicarbonate indicator. pH was not measured for the light/dark shift experiments with algae bead bracelets. For testing the effect of different dark exposure times on photosynthesis, one algae bead bracelet was exposed to 9 h of darkness and the other was exposed to 15 h of darkness. After the dark exposure, the 9 h- and 15 h- dark-adapted bracelets were exposed to light for 4 h and 12 h, respectively and were imaged after the light exposure. pH of the water inside these light and dark-exposed bracelets was measured using pH testing strips (Fisher Scientific, Waltham, MA).

Light and dark exposure experiments with Chlamydomonas strain-specific beads in glass vials

For testing the effect of water quality on photosynthesis, eight 4A+ beads were either introduced into 2.5 mL of tap water (pH 7.2–7.3) or into de-ionized [DI] water (pH 7.1–7.2) in 5.5 mL screw capped glass vials (Fisher Scientific, Waltham, MA). For testing the effect of cell density on photosynthesis, eight 4A+ beads were either introduced into 2.5 mL of tap water (pH 7.2–7.3). For both stated experiments, 125 µL of the 0.02% phenol red solution (Fisher Scientific, Waltham, MA) was added to the algae bead vials to serve as a pH indicator and the vials were capped tightly. One set of 4A+ bead and the control vials were exposed to 150–200 µmole m−2s−1 light intensity and the other set to darkness for 2 h. After 2 h of light or dark exposure, vials were imaged and pH of the water in the vials was measured using a Thermo Fisher Scientific Orion-3 Star benchtop pH meter (Fisher Scientific, Waltham, MA).

For comparative analyses of photosynthesis and cellular respiration in 4A+ and 10E35 strains under constant light/darkness, beads having approximately 2 × 106 cells/bead for each strain were used (Fig. S2). Eight 4A+ and 10E35 beads were introduced into 2.5 mL of tap water (pH 6.9–7.3) in 5.5 mL screw capped glass vials (Fisher Scientific, Waltham, MA). 125 µL of the phenol red solution (Fisher Scientific, Waltham, MA) was added to the 4A+ and 10E35 bead vials and the vials were capped tightly. One set of 4A+, 10E35 and control vials was exposed to light intensity of 150–200 µmole m−2 s−1 and the other set was exposed to darkness for 1 h. The algae bead and control vials were imaged after every 30 min over a period of 1 h and pH of the water in the vials was measured. The 1-h light adapted 4A+, 10E35 and the control vials were exposed to light for an additional 3 h and then shifted to dark. The vials were imaged after every 15 min for a period of 1 h during dark exposure. After 1 h, these dark-exposed vials were kept under dark for additional 5 h. After 6 h-of dark exposure, vials were shifted to light (150–200 µmole m−2 s−1) and imaged after 30 min, 1 h, 2 h, 3 h and 48 h. pH of the water in the glass vials in the above stated experiments were measured using a Thermo Fisher Scientific Orion-3 Star benchtop pH meter (Fisher Scientific, Waltham, MA).

Imaging and Data analyses

Images were captured a Samsung Galaxy S5 camera. Statistical analyses of the recorded pH data were performed using Microsoft Excels’ t-Test: Paired Two Sample for Means tool in the analysis ToolPak. Both One-Tailed and Two-Tailed Hypothesis Tests were performed. Standard deviations shown in Tables under result section was calculated using Excel. Raw statistical analyses data from three biological replicates per experiment have been deposited in Figshare (https://doi.org/10.6084/m9.figshare.12344024.v1) and are presented in the Data S1. p-values of experiments can be found in the Data S1. Data S2 contains raw pH data, mean and standard deviation information. Each biological replicate had three internal replicates. The average of three internal replicates from each biological replicate is shown in the data in Data S1 and S2.

Results

Photosynthesis and cellular respiration-induced pH changes in de-ionized (DI) water and tap water vials containing Chlamydomonas 4A+ strain beads

DI water is known to contain less dissolved gases and minerals than tap water (The Lab Depot, 2020; Whitehead, 2020). Amounts of dissolved oxygen and carbon-dioxide in the water used for photosynthesis monitoring experiment will affect the results in a photosynthesis lab. Hence, we monitored photosynthesis and cellular respiration of the wild type Chlamydomonas strain, 4A+ beads in DI water and in tap water to see which one would be suitable for designing the photosynthesis lab (Fig. 1; Table S1). The pH indicator phenol red exhibits a gradual transition from light orange to red over the pH range 6.8 to 8.2. Phenol red turns yellow below pH 6.7 and turns to a bright pink (fuchsia) color above pH 8.2. The expected color scale at different pH when phenol red is used as the pH indicator can be found at https://commons.wikimedia.org/wiki/File:Phenol_red_pH_6,0_-_8,0.jpg and at https://en.wikipedia.org/wiki/Phenol red.

Figure 1 Photosynthesis and cellular respiration-induced pH/color changes in vials containing Chlamydomonas 4A+ strain beads in de-ionized water and tap water.

C stands for control vials which do not contain algae beads and E stands for experimental vials containing algae beads. (A) Color changes in de-ionized water. (B) Color changes in tap water. Vials were exposed to light and darkness for 2 hours. Algal beads had approximately 2 × 106 cells/beads and 8 algal beads were used per experimental vial. All statistical analyses can be found in https://doi.org/10.6084/m9.figshare.12344024.v1, Data S1 and S2 and Table S1.

Our results show that there was no statistically significant difference in the water color or pH between the control and experimental vials containing de-ionized water in the light and in the dark (p-values from the 1-tailed and 2-tailed hypothesis tests for the light set samples were 0.21 and 0.42, respectively; p-values from the 1-tailed and 2-tailed hypothesis tests for the dark set samples were 0.09 and 0.18, respectively; Fig. 1A; https://doi.org/10.6084/m9.figshare.12344024.v1; Data S1 and S2). There was no significant difference in pH in the control vials with DI and tap water under light and dark (p-values for the control light samples from the 1-tailed and 2-tailed hypothesis tests were 0.11 and 0.22, respectively; p-values for the control dark samples from the 1-tailed and 2-tailed hypothesis tests were 0.09 and 0.18, respectively; Data S1). Dark-exposed tap water experimental vial displayed acidic pH (yellow color) while the light-exposed tap water experimental vial displayed alkaline pH (fuchsia color) relative to the respective control vials and the pH difference was statistically significant (p-values from the 1-tailed and 2-tailed hypothesis tests were 0.0001 and 0.0002, respectively; Fig. 1B; Table S1; https://doi.org/10.6084/m9.figshare.12344024.v1; Data S1 and S2). There was a difference of approximately 2 pH units between the dark and light-adapted tap water experimental vials in contrast to the 0.07 pH unit difference between the in the dark-and in the light-adapted DI water experimental vials (Table S1; https://doi.org/10.6084/m9.figshare.12344024.v1; Data S1 and S2). In the light-exposed tap water algal bead vial, buoyancy of a bead can be seen which is indicative of O2 production in photosynthesis (Fig. 1B). Taken together our results show that tap water is preferable over DI water for performing photosynthesis lab activities.

Figure 2 Effect of cell numbers per Chlamydomonas 4A+ strain bead on photosynthesis and cellular respiration-induced pH/color changes in tap water.

“C” stands for control vials that do not contain algae beads. “E” stands for experimental vials containing algae beads. (A) Color changes in experimental vials that contained beads which had approximately 4 × 106 cells/bead. (B) Color changes in experimental vials that contained beads which had approximately 2 ×106 cells/bead. Light and dark exposure of vials was for 2 hours. Eight algal beads were used per experimental vial. All statistical analyses can be found in https://doi.org/10.6084/m9.figshare.12344024.v1, Data S1 and S2 and Table S2.

Effect of total cell numbers in Chlamydomonas 4A+ strain beads on photosynthesis and cellular respiration-induced color/pH changes in tap water

We used two types of beads that have two-fold difference in total cell numbers/bead: (1) beads that have approximately 2 × 106 cells/bead and, (2) beads that have approximately 4 × 106 cells/bead. It is expected that a high cell number in a bead will increase cellular respiration as a high cell density in the bead will create oxygen stress. The pH in the light-exposed vial containing 4 × 106 cells/bead was 6.1 and the pH in the light exposed vial containing 2 × 106 cells/bead was pH 8.4 for the same duration of light exposure (Fig. 2; Table S2; https://doi.org/10.6084/m9.figshare.12344024.v1; Data S1 and S2). The pH in the light- and dark-exposed vials containing 4 × 106 cells/bead differed by only 0.1 pH unit while the pH in the light- and dark-exposed vials containing 2 × 106 cells/bead differed approximately by 2 pH units for the same duration of light exposure (Fig. 2; Table S2; https://doi.org/10.6084/m9.figshare.12344024.v1; Data S1 and S2). There was a statistically significant pH difference between the light-exposed vials with 4 × 106 cells/bead and that with 2 × 106 cells/bead (p-values from the 1-tailed and 2-tailed hypothesis tests for the light-exposed vials were 0.0001 and 0.0002, respectively; Data S1). The pH difference between the dark-exposed vials with 4 × 106 cells/bead and that with 2 × 106 cells/bead was statistically significant (p-values from the 1-tailed and 2-tailed hypothesis tests were 0.01 and 0.02, respectively; Data S1). pH differences between the light and dark control vials were insignificant as the p-values were higher than 0.05 in both 1-tailed and 2-tailed hypothesis tests (Data S1). Our results show high cell density/bead will hinder observation of pH changes in a photosynthesis lab. In the light-exposed experimental vial, partial buoyancy of one bead can be seen, indicative of O2 production in photosynthesis (Fig. 2B).

Indirect detection of carbon dioxide concentration in the 4A+ bead bracelet under light and darkness using the bicarbonate indicator

Bicarbonate indicator is commonly used in photosynthesis and respiration experiments to detect indirectly the percentage of carbon dioxide in a sample. It is a more sensitive pH indicator than phenol red. When the carbon dioxide content in water is higher than 0.04%, pH becomes acidic. Acidic pH changes the red color of the indicator to yellow. If the carbon dioxide content is lower than 0.04%, pH gets alkaline and the indicator changes color from red to magenta and, under very low carbon dioxide concentrations the color of the indicator changes to purple (https://en.wikipedia.org/wiki/Bicarbonate_indicator). The expected color scale at different pH when bicarbonate indicator is used as the pH indicator can be found at https://pmgbiology.com/tag/respiration/.

We used three bracelets (with algal beads ranging from 30–38) to monitor color changes of the bracelet water containing the bicarbonate indicator. These are designated as control, dark-exposed and light-exposed bracelets in Fig. 3. The color of the water in the control bracelet (not exposed to dark or light), dark- and light-exposed bracelets were, light red, bright yellow and dark blue, respectively (Fig. 3). The objective of the experiment was to simply determine the color/pH changes of the water in the experimental bracelets in the light or in the dark relative to the control. The average pH of the water in the control algal bracelet was around 7 (Fig. 3A; Table S3; https://doi.org/10.6084/m9.figshare.12344024.v1; Data S1 and S2). pH of the water in the dark-exposed algal bracelets ranged between 6 and 6.5 with STDEV ± 0.24 (Fig. 3B; https://doi.org/10.6084/m9.figshare.12344024.v1; Data S1 and S2) indicating a high percentage of carbon dioxide because of cellular respiration. pH of the water in the light-exposed algal bracelets ranged between 8.5 and 9 with STDEV ± 0.24 indicating a low percentage of carbon dioxide because of photosynthesis (Fig. 3C; Table S3; https://doi.org/10.6084/m9.figshare.12344024.v1; Data S1 and S2). pH difference between the light- and dark-exposed bracelets is statistically significant (p-values from the 1-tailed and 2-tailed hypothesis tests were 0.001 and 0.003, respectively; Data S1 and S2; https://doi.org/10.6084/m9.figshare.12344024.v1). Our results clearly show that carbon-dioxide percentage can be monitored indirectly under light/darkness via sharp pH/color changes in the water in the bracelet in the presence of the bicarbonate indicator.

Figure 3 Indirect monitoring of carbon dioxide percentage inside the 4A+ bead bracelet under constant light and darkness using bicarbonate indicator.

(A) Control algal bracelet that has not been exposed to light or to darkness (zero time point). (B) An algal bracelet that was exposed to darkness for 3 hours. (C) An algal bracelet that was exposed to light for 3 hours. Water color change was monitored in dark and light. Algal beads had approximately 2 × 106 cells/bead. Number of algal beads in the control, dark- and light-exposed bracelets were 32, 38 and 38, respectively. All statistical analyses can be found in https://doi.org/10.6084/m9.figshare.12344024.v1, Data S1 and S2 and Table S3.

Time course monitoring of photosynthesis-induced color changes in the dark adapted-4A+ bead bracelet when shifted to light

An algal bracelet was dark-adapted for 4 h. After dark-adaptation the bracelet was exposed to light for 4 h. This light-exposed bracelet was imaged after every 1 h during light exposure to monitor the gradient color changes over time without disturbing the bracelet (Fig. 4). The results show that if the algal bracelet is left undisturbed, one can pinpoint specifically which beads were actively photosynthesizing from the red-magenta-purple color streaks in the water on top of these beads that were removing carbon-dioxide from the tubing water (Figs. 4B–4D). pH was not measured in these bracelets as the objective of this experiment was to determine if differences exist in photosynthetic rates among different beads by visually observing the gradual color change of the water in the bracelet in light.

Figure 4 Time course monitoring of photosynthesis-induced color/pH changes in the dark-adapted 4A+ bead bracelet when shifted to light.

(A) An algal bracelet that was dark adapted for 4 hours. (B) Dark-adapted bracelet exposed to light for 1 hour. (C) Dark-adapted bracelet exposed to light for 2 hours. (D) Dark-adapted bracelet exposed to light for 3 hours. (E) Dark-adapted bracelet exposed to light for 4 hours. Algal beads have approximately 2 × 106 cells/bead. Number of algal beads in the bracelet was 15. Bicarbonate indicator was used as a pH indicator.

Time course monitoring of cellular respiration-induced color changes in the light adapted-4A+ bead bracelet when shifted to darkness

An algal bracelet was light adapted for 4 h. After light-adaptation the bracelet was exposed to dark for 3 h. The dark-exposed bracelet was imaged after every 1 h during the light exposure to monitor the gradient color changes over time (Fig. 5). The results show that distinct pH gradient can be observed in a colorful way in an undisturbed algal bracelet (Fig. 5B). pH was not measured in these bracelets as the objective of this experiment (in conjunction with the Fig. 4 experiment) was to teach students in a fun way, the “tug of war” between photosynthesis and cellular respiration by visually observing the dramatic color changes of the water in the bracelet upon exposure to darkness or light.

Figure 5 Time course monitoring of cellular respiration-induced color/pH changes in the light-adapted 4A+ bead bracelet when shifted to darkness.

(A) An algal bracelet that was light adapted for 4 hours. (B) Light-adapted bracelet exposed to dark for 1 hour. (C) Light-adapted bracelet exposed to dark for 2 hours. (D) Light-adapted bracelet exposed to dark for 3 hours. Algal beads had approximately 2 × 106 cells/bead. The bracelet contained thirty-six 4A+ strain beads. Bicarbonate indicator was used as a pH indicator.

Effect of prior dark exposure duration on photosynthesis-induced pH changes in the 4A+ bead bracelet in light

One algal bracelet was kept in the dark for 9 h and the other one was kept in the dark for 15 h. After dark incubation, both bracelets were imaged and the pH was measured using pH testing strips and, then shifted to light (Fig. 6). There was no significant difference in pH between the 9-h dark-adapted and 15-h-dark adapted bracelets (p-values from 1-tailed and 2-tailed tests were 0.11 and 0.22, respectively; Data S1). There was a significant difference in pH between 9-h-dark adapted bracelet and 15-h-dark adapted bracelet when these were exposed to light for 4 h and 12 h, respectively (p-values from 1-tailed and 2-tailed tests were 0.002 and 0.005, respectively; Data S1). The bracelet that was kept in dark for 9 h showed increase in pH from pH 6 [STDEV ± 0] to pH 8.67 [STDEV ± 0.24] within 4 h under light because of photosynthesis (Figs. 6A & 6B; Table S4; https://doi.org/10.6084/m9.figshare.12344024.v1; Data S1 and S2). The bracelet that was kept in dark for 15 h showed a small increase in pH from 5.5 [STDEV ± 0.41] to 6.3 [STDEV ± 0.24], despite being exposed to light for 12 h. This indicates prior prolonged exposure to darkness hinders photosynthesis in algal beads in light (Figs. 6C & 6D; Table S4; https://doi.org/10.6084/m9.figshare.12344024.v1; Data S1 and S2).

Figure 6 Effect of prior dark exposure duration on photosynthesis-induced color/pH changes in 4A+ bead bracelet in light.

(A) An algal bracelet that was dark adapted for 9 hours. (B) 9 hours-dark adapted-bracelet shifted to light for 4 hours. (C) An algal bracelet that was dark adapted for 15 hours. (D) 15 hours-dark adapted-bracelet shifted to light for 12 hours. Algal beads had approximately 2 ×106 cells/bead. Both bracelets contained 38 beads. All statistical analyses can be found in https://doi.org/10.6084/m9.figshare.12344024.v1, Data S1 and S2 and Table S4.

Comparative studies of photosynthesis and cellular respiration- induced color/pH changes in vials containing wild type 4A+ and 10E35 mutant beads

4A+ and 10E35 beads have approximately 2 × 106 cells/bead (Fig. S2). Each light and dark set comprised of a control and experimental vials of 10E35 and 4A+ (Fig. 7). Images of the vials in each light and dark set were taken before light or dark exposure (Figs. 7A and 7D). Each light and dark vial sets were imaged after 30 min of light and dark exposures, respectively for a period of 1 h. Results show a statistically significant slow increase in pH in 10E35 vial under light compared to that in the 4A+ vial (Figs. 7B and 7C; Table S5; https://doi.org/10.6084/m9.figshare.12344024.v1; Data S1 and S2) (p-values from the 1-tailed and the 2-tailed tests ranged from 0–0.015 and 0–0.03, respectively). This could be due to a slow rate of photosynthesis or a high rate of cellular respiration or a combination of both phenomena in 10E35 relative to that in 4A+. 10E35 displays relatively a higher rate of cellular respiration in dark compared to that in 4A+ as indicated by the fast pH drop in dark in 10E35 vial over time compared to that in the 4A+ vial that is statistically significant (Figs. 7E–7F; Table S5, https://doi.org/10.6084/m9.figshare.12344024.v1; Data S1 and S2) (p-values from the 1-tailed and 2-tailed tests ranged from 0.002–0.011 and 0.005–0.022, respectively).

Figure 7 Comparative studies of photosynthesis and cellular respiration-induced pH/color changes in wild type 4A+ and 10E35 bead vials under light and darkness.

(A) Control, 10E35 and 4A+ bead vials before light exposure. (B) Control, 10E35 and 4A+ bead vials after 30 minutes of light exposure. (C) Control, 10E35 and 4A+ bead vials after 1 hour of light exposure. (D) Control, 10E35 and 4A+ bead vials before dark exposure. (E) Control, 10E35 and 4A+ bead vials after 30 minutes of dark exposure. (F) Control, 10E35 and 4A+ bead vials after 1 hour of dark exposure. Algal beads of each strain had approximately 2 × 106 cells/bead. Eight beads of each strain were used per experimental vial for the experiment. All statistical analyses can be found in https://doi.org/10.6084/m9.figshare.12344024.v1, Data S1 and S2 and Table S5.

Time course monitoring of cellular respiration-induced color/pH changes in the dark in vials containing 4A+ and 10E35 beads that were exposed to light for 4 h

Light-exposed 10E35, 4A+ and control vials from Fig. 7 experiment were exposed to light for additional 3 h. Hence this set was light-exposed for a total of 4 h. After fours of light exposure, images were taken and the vials were exposed to dark. Images of the dark-exposed vials were taken every 15 min over a period of 1 h during dark exposure (Fig. 8). 10E35 shows relatively a higher cellular respiration rate that is statistically significant, compared to that in 4A+, as indicated by the rapid drop in pH in the 10E35 vial compared to that in the 4A+ vial (p-values from the 1-tailed and 2-tailed tests ranged from 0.0005–0.001 and from 0.001–0.003, respectively) (Figs. 8B–8E; Table S6; https://doi.org/10.6084/m9.figshare.12344024.v1; Data S1 and Data S2). The results re-confirm the results shown in Figs. 7E–7F (Table S5, https://doi.org/10.6084/m9.figshare.12344024.v1; Data S1 and Data S2).

Figure 8 Time course monitoring of cellular respiration-induced color/pH changes in the dark in 4A+ and 10E35 bead vials that were adapted to light for 4 hours.

These vials are the light-adapted vials from the Fig. 7 experiments. (A) Light-adapted control, 10E35 and 4A+ bead vials before dark exposure. (B) Control, 10E35 and 4A+ bead vials after 15 minutes of dark exposure. (C) Control, 10E35 and 4A+ bead vials after 30 minutes of dark exposure. (D) Control, 10E35 and 4A+ bead vials after 45 minutes of dark exposure. (E) Control, 10E35 and 4A+ bead vials after 1 hour of dark exposure. All statistical analyses can be found in https://doi.org/10.6084/m9.figshare.12344024.v1, Data S1 and S2 and Table S6.

Time course monitoring of photosynthesis-induced pH changes in the light in 4A+ and 10E35 bead vials that were exposed to dark for 6 h

Dark-exposed 10E35, 4A+ and control vials from Fig. 8 experiment were exposed to dark for additional 5 h. Hence this set was dark-exposed for a total of 6 h. After 6 h of dark exposure, images were taken of the dark-exposed vials and the vials were exposed to light. Images of these light-exposed vials were taken after 30 min, 1 h, 2 h, 3 h and 48 h of light exposure (Fig. 9). The results show that 4A+ photosynthesized at a faster rate compared to 10E35 after 6 h of dark exposure to cause a distinct water color/pH change that was statistically significant (p-values from the 1-tailed and 2-tailed tests ranged from 0.0001–0.0175 and from 0.0002–0.035, respectively) (Table S7; https://doi.org/10.6084/m9.figshare.12344024.v1; Data S1 and S2). Despite the significant pH difference in the 6 h dark-adapted and 48 h light-adapted vials of 10E35, pH in the 48 h-light exposed 10E35 vial was acidic (pH = 6.43 ± 0.06) compared to the alkaline pH (8.47 ± 0.06) in the 48 h-exposed 4A+ vial (Table S7; https://doi.org/10.6084/m9.figshare.12344024.v1; Data S1 and S2). It is known that 10E35 progressively photo-bleaches with increase in light intensity (Nguyen et al., 2017; Article S2). Figure 9F shows that 10E35 beads were photo-bleached in the light-exposed vial after 48 h of light exposure. Photo-bleaching indicates that there is chlorophyll breakdown in the beads.

Figure 9 Time course monitoring of photosynthesis-induced color/pH changes in the light in 4A+ and 10E35 bead vials that were adapted to darkness for 6 hours.

These vials are the dark-adapted vials from the Fig. 8 experiments. (A) Dark-adapted control, 10E35 and 4A+ bead vials before light exposure. (B) Control, 10E35 and 4A+ bead vials after 30 minutes of light exposure. (C) Control, 10E35 and 4A+ bead vials after 1 hour of light exposure. (D) Control, 10E35 and 4A+ bead vials after 2 hours of light exposure. (E) Control, 10E35 and 4A+ bead vials after 3 hours of light exposure. (F) Control, 10E35 and 4A+ bead vials after 48 hours of light exposure. All statistical analyses can be found in https://doi.org/10.6084/m9.figshare.12344024.v1, Data S1 and S2 and Table S7.

Discussion

Chlamydomonas reinhardtii is a unicellular micro-green alga (a Chlorophyte) that retains many of the features of green plants and of the common ancestor of plants and animals, although its lineage diverged from Streptophytes over one billion years ago. Chlamydomonas is used to study eukaryotic photosynthesis because, unlike angiosperms, it can use acetate to grow in the dark while maintaining a functional photosynthetic apparatus (Merchant et al., 2007). It is also a model organism for elucidating eukaryotic flagella and basal body structure and functions which can be linked to various ciliopathies (Silflow & Lefebvre, 2001). More recently, Chlamydomonas research has been developed for bioremediation purposes, generation of biofuels and has led to breakthroughs in optogenetics (Merchant et al., 2007; Critical tool for brain research derived from “pond scum” NSF, 2013; Zhang, 2015; Scranton et al., 2015).

Currently, the Chlamydomonas Resource Center [https://www.chlamycollection.org/ ] offers number of educational kits (Resources For Teaching—Chlamydomonas Resource Center) including instructions and strains on its website; however, these tools barely scratch the surface of what could be taught using Chlamydomonas to students enrolled in K12 Biology and in college Biology undergraduate courses. Hence there is a huge potential to develop Chlamydomonas an under-utilized teaching tool, into a powerful popular teaching tool which will complement existing plant science teaching strategies. Our objective for the American Society of Plant Biologists’ (ASPB) Plant-BLOOME project was to design fifteen simple hands-on activities on different Biology topics that can not only educate and excite high school students about Chlamydomonas but can be also included as a component in college Biology laboratory courses. The activities described in this manuscript are centered on photosynthesis and cellular respiration.

We have found that the Chlamydomonas culture should be grown under low light (80–100 micro mol photons m−2 s−1) to obtain a healthy culture that is not photo-oxidatively stressed, to be used for our lab activities. The culture should be a dense culture and have a cell density ranging from 18 × 106 cells/mL to 22 × 106 cells/mL to get enough cells for a class of 24 students, working in groups of two to three. Algal beads once made, should be rinsed thoroughly with tap water for at least 5 min to remove residual sodium chloride that is formed as a product in the reaction between sodium alginate and calcium chloride during bead-making step. This step is a very important step and must not be skipped as any residual sodium chloride will hinder photosynthesis in the experiment. As shown in the Fig. 2, high cell numbers in a bead has a negative effect on photosynthesis. We have found best results can be achieved when the harvested cell numbers are between 250 × 106 − 395 × 106 cells/ 50 mL falcon tube. Cells inside the beads are oxygen-stressed. Hence it is important to leave air gaps inside the bracelet at each end of the flexible tubing (see Materials and Methods). The same rule applies when performing the experiment in a 5.5 mL glass vial. It is important to leave air gap of half the volume of the vial.

Results in Fig. 6 showed that prolonged dark exposure of 15 h has a negative effect on photosynthesis. Algae bead bracelets exposed to 9 h of darkness (Fig. 6) can be shifted back and forth between dark and light to display color changes over a period of 24–48 h (the color changes slowly after 24 h; based on observations in different classrooms, no data was collected). We have also found that once the bracelet is assembled, if it is exposed to light for about 3–4 h (which we call in our lab as the “light charging of the bracelet”) and, then switched to dark for 2–3 h [“discharging of the bracelet”], the bracelet displays fast color changes as long as the dark exposure time was not exceeded beyond 9 h (based on observations in different classrooms, no data was collected). But if the assembled bracelet is shifted to dark immediately after assembly, the bracelet fails to display fast color changes. Immobilized oxygen stressed-Chlamydomonas cells in the beads are dependent on photosynthesis for glucose biosynthesis as they are immersed in tap water in the bracelet. Tap water lacks acetate and other nutrients for algal growth. We hypothesize that the initial light exposure allows the cells to synthesize glucose/starch by photosynthesis, which is later used to support the high rate of cellular respiration in beads for energy production. If the bracelet is shifted to dark without prior light exposure, the high cellular respiration rate consumes the existing starch in the cells. Hence when this dark-exposed bracelet is exposed to light, cells will have to synthesize enough glucose/starch via photosynthesis to support the high rate of cellular respiration in the beads and this will take some time. This is reflected in the slow color/pH changes of a bracelet that is exposed to dark immediately after assembly compared to the one that is exposed to light immediately after assembly.

Chlamydomonas can take up exogenous acetate from the TAP medium to make net synthesis of glucose via the Glyoxylate/C2 cycle, which is present in many bacteria, micro-algae and plants (Kunze et al., 2006). Substituting tap water with acetate containing-TAP growth media (heterotrophic and photo-heterotrophic media; Text S1) inside the bracelet will hinder color change in bracelets/vials as TAP medium has Tris buffer, which has a pKa value of 8.06 at 25 °C and a buffering range of pH 7–9. Chlamydomonas grows slower in High Salt (HS) photosynthetic media than in TAP medium as HS medium lacks acetate (Sueoka, 1960). HS medium cannot be used as a substitute for tap water inside the bracelet as we have tried it and have found that the bracelets do not show color changes even if the beads are exposed to light for 48 h (data not shown).

During spring 2018-fall 2019, the described laboratory activities were incorporated in Biology classes in nine schools and in Biology labs at the University of West Georgia and at the Perimeter College [Georgia State University] (Text S1). To date, we have targeted of about 947 school students in Georgia and hope to target more college students in future. We are proposing a class workflow in Text S3, which is based on the feedback of 12 school teachers and 2 college instructors who participated in the Plant-BLOOME project. Regardless of the suggested time line, instructors can adjust lab times according to their teaching agenda by either spreading the lab activities across multiple classes or by removing one or more activities (Text S3). This will allow the instructor to involve the class in discussion after each activity. Alternatively, students can perform an outdoor experiment by wearing these bracelets/necklaces (you can also make algae bead necklaces) during day time and exposing these bracelets to strong sunlight or wear them in the night to see the water color changes. Conducting the experiment in a 5.5 mL capped glass vials will expedite the experiment completion within 1.5–2 h in classrooms. The advantage of performing the experiment in glass vials is that students can clearly monitor oxygen production in photosynthesis by monitoring the buoyancy of the algal beads over time. Bead buoyancy is difficult to clearly visualize in a bracelet because of the narrow diameter of the bracelet tubing.

Table 1 Customization of the photosynthesis lab for middle school students.

Activities	Comments	
1. Observation of swimming Chlamydomonas cells under bright light microscope	Students can observe the orange eye spot that functions like “human eye” and helps in photo-taxis. Orange color of eyespot is due to carotenoids.	
2. Chlamydomonas photosynthetic pigment analyses by paper chromatography	Pigments can be extracted by teachers using 100% acetone from pigment-deficient photosynthetic strains and wild type strains. Spinach extract and cyanobacterial Phycocyanin powder (Fisher Scientific) can be used in the paper chromatography experiment to compare pigments in phylogenetically separated species. Chromatography lab can be also connected with fall leaf color change biochemistry and carotenoids.	
3. Students can observe the red fluorescence of Phycocyanin solution under cell phone LED light	Students will learn the oceans can change in color in summer and in winter because of cyanobacterial bloom concentration and will know about cyanobacterial pigments: phycocyanin and phycoerythrin.	
4. Photo-taxis lab using Chlamydomonas wild type strain and the lts1-211 mutant (Ueki et al., 2016)	lts1-211 is an eye spot-less mutant and is devoid of carotenoids that shows reverse phototactic behavior compared to the wild type strain in the light. Students can point a flash light to culture plates of the two strains and observe the phototaxis pattern of the two strains. lts1-211 moves towards light while the wild type strain moves away from the light.	
5. Photosynthesis and cellular respiration labs	As described in this manuscript	

Table 2 Customization of the photosynthesis lab for high school students.

Activities	Comments	
1. Observation of swimming Chlamydomonas cells under bright light microscope	Students can observe the orange eye spot that functions like “human eye” and helps in photo-taxis. Orange color of eyespot is due to carotenoids.	
2. Chlamydomonas photosynthetic pigment analyses by paper chromatography	Pigments can be extracted by teachers using 100% acetone from pigment-deficient photosynthetic strains and wild type strains. Spinach extract and cyanobacterial Phycocyanin powder (Fisher Scientific) can be used in the paper chromatography experiment to compare pigments in phylogenetically separated species. Chromatography lab can be also connected with fall leaf color change biochemistry and carotenoids.	
3. Students can observe the red fluorescence of Phycocyanin solution under cell phone LED light	Students will learn the oceans can change in color in summer and in winter because of cyanobacterial bloom concentration and will know about cyanobacterial pigments: phycocyanin and phycoerythrin.	
4. Photo-taxis lab using Chlamydomonas wild type strain, lts1-211 mutant and rescued lts1-211 (Ueki et al., 2016).	lts1-211 is an eye spot-less mutant and is devoid of carotenoids that shows reverse photo-tactic behavior compared to the wild type strain in the light. Students can point a flash light to culture plates of the two strains and observe the phototaxis pattern of the two strains. lts1-211 moves towards light while the wild type strain moves away from the light. Rescued lts1-211 will show similar photo-tactic behavior as the wild type strain	
5. Observation of Phototaxis under different light intensities and redox conditions.	lts1-211 mutant moves away from reactive oxygen species like hydrogen peroxide unlike the wild type strain.	
6. Bioinformatics lab	As described under the Discussion section in the manuscript.	
7. Photosynthesis and cellular respiration labs	As described in this manuscript	
8. Advanced Photosynthesis Labs: Testing different light intensities and different light color using colored filters (red, blue and green)	As described under the Discussion section in the manuscript.	

Table 3 Customization of the photosynthesis lab for college undergraduates.

Activities	Comments	
1. Comparative photosynthesis studies of a wild type and a photosynthetic mutant strain using strain-specific algae beads in vials.	Same type of experiment as that described for 4A+ and 10E35 in the article.	
2. PCR using mutated gene-specific primer using genomic DNA of the mutant and the wild type strain	Students learn how to isolate genomic DNA	
3. DNA gel electrophoresis and agarose gel extraction of the PCR product	Students learn molecular techniques	
4. Cloning of the gel extracted PCR product, DNA sequencing of the clone and analyses of DNA sequencing data	Students learn molecular techniques	
5. Western blotting to detect presence or absence of the protein in the wild type and mutant strains	If the protein-specific antibody is available.	
6. Bioinformatic labs	Using various web-based free programs to perform multi-sequence alignments of DNA/protein sequences and generating phylogenetic trees, identifying conserved domains, studying gene expression and gene co-expression and generating gene network; prediction of protein location in cell, learning to use different gene/protein databases etc.	
7. Photosynthetic pigment analyses by pigment extraction and spectrophotometry; Paper chromatography or thin layer chromatography-based labs.	Many photosynthetic mutants are deficient in chlorophyll and carotenoids. Chlorophyll and carotenoids can be extracted by 100% acetone.	
8. Photosynthesis and Non-photochemical quenching studies using sophisticated equipment.	Only institutions that have an oxygen electrode and a PAM fluorometer can perform these activities.	

Photosynthetic efficiencies of Chlamydomonas strains are measured in a laboratory by an oxygen electrode. But many financially disadvantaged schools and institutions of higher learning do not have access to an oxygen electrode. Our hands-on activity can be used to compare crudely photosynthetic efficiencies of Chlamydomonas wild type and photosynthetic mutant strains in a classroom setting. This will allow educators at institutions with limited resources and funding to engage students in critical thinking based on observations of a scientific experiment.

Our lab activities can be customized for different grade levels by adding or removing layers of lab components. Some suggested activities for middle school, high school and college undergraduates are shown in Tables 1, 2 and 3, respectively. For example, for middle school students the algae bead bracelet or the vial version of the experiment can be used. Students can observe under light microscopes, swimming Chlamydomonas and its bright orange eyespot which is used by the cell for light sensing and aids photo-taxis (Table 1; Ueki et al., 2016). Photosynthesis is modulated by light color and light intensities (Tymoczko, Berg & Stryer, 2015). Red and blue light stimulates photosynthesis and other colored light are not utilized for photosynthesis (Tymoczko, Berg & Stryer, 2015). Hence algae bead bracelets can be used by high school students to test the effects of different light intensities and colored light using different colored light filters (Table 2).

High school students can also conduct a vial experiment with a wild type strain and any available photosynthetic mutant strain that they have access to. Photosynthetic mutants like the cytochrome f deficient mutant (ΔpetA) [CC-3737 petA (N153Q)]; the D1-less mutant (Fud7) [CC-4147 FUD7 (psbA deletion) mt+] and the D2-less mutant (ΔPsbD) [CC-4385 PsbD (deletion) mt+ are available via Chlamydomonas Resource Center (Table 2). 10E35 mutant can be obtained from our lab. Additionally, a basic bioinformatic laboratory can be added to the high school Biology lab. The DNA sequence of the mutated gene in the photosynthetic Chlamydomonas mutant can be given to students and they can use the DNA sequence to BLAST the NCBI database to identify the gene and the protein. Students can also check for paralogs/orthologs of the identified gene/protein (Table 2). For college undergraduate level Biology labs, additional molecular and biochemical layers can be added on top of the high school lab components as shown in Table 3.

We have provided class work-flow, sample pre- and post-lab questions and a rubric for grading pre- and post-lab assignments which can be used by educators (Text S3). The assignments and the rubric can be customized according to the knowledge base of students in the class. In summary, science literacy in young students can be improved by studying a “pond-scum” which is used by plant biologists, neuroscientists, biomedical and renewable energy researchers and can show them the inter-disciplinary nature of 21st century Biology.

Conclusions

Our designed protocol can be used to make beads using motile micro-alga like Chlamydomonas reinhardtii. These algal beads can be used for basic photosynthesis labs or for comparative studies of relative rates of photosynthesis and cellular respiration in Chlamydomonas wild type and mutant strains. Although our work was performed with the objective of designing engaging hands-on plant biology labs for K16 Biology students, it might be useful to bioenergy researchers who are interested in exploring the use of immobilized Chlamydomonas or other motile green algae for biofuel production (Scranton et al., 2015; Radakovits et al., 2010; Canbay, Kose & Oncel, 2018). Our lab activities using the wild type Chlamydomonas strain can be performed both in glass vials and in bracelets. Based on our class room experiences at nine schools and two colleges in Georgia and the enthusiasm of the plant community members at the educational booths at the Plant Biology meetings organized by ASPB, we envision that young students will find the ‘bracelet’ approach more enjoyable than conducting the same experiment in glass vials (Text S1). Our lab activities are inexpensive and can been customized according to grade levels.

Supplemental Information

Supplemental Information 1 Glimpses of dissemination of our educational outreach Plant-BLOOME project

Click here for additional data file.

Supplemental Information 2 Lab supply items, prices, cost comparison and TAP medium recipe

Click here for additional data file.

Supplemental Information 3 Teaching accessories

Click here for additional data file.

Figure S1 Shape diversities of Chlamydomonas beads due to non-uniform pipetting and improper mixing algal cell-sodium alginate suspension

When algal cell-sodium alginate suspension is not well mixed in between pipetting and, pipetting is not performed in a steady uniform fashion, beads will assume different shapes and color because of different numbers of cells in the cell suspension drop.

Click here for additional data file.

Figure S2 4A+ and 10E35 beads with approximately equal cell numbers per bead

(A) 4A+ beads. (B) 10E35 beads. The beads of each strain have approximately 2 × 106 cells/ bead. Note: 10E35 has less chlorophyll per cell compared to 4A+ (Nguyen et al., 2017, Article S2). Hence 10E35 beads appear slightly less dark green than the 4A+ beads.

Click here for additional data file.

Data S1 pH data with statistical analyses from experiments under light and darkness using Chlamydomonas reinhardtii bead bracelets and glass vials containing Chlamydomonas beads

Data S1 contains raw pH data with statistical analyses for experiments using Chlamydomonas beads, described in the following figures: Fig. 1, Fig. 2, Fig. 3, Fig. 6, Fig. 7, Fig. 8 & Fig. 9. Data analyses is based on three biological replicates. Each biological replicate had three internal replicates.

Click here for additional data file.

Data S2 pH raw data with means and standard deviations

Data S2 file contains the raw pH data with mean and standard deviation information from the experiments described in the following figures: Fig. 1, Fig. 2, Fig. 3, Fig. 6, Fig. 7, Fig. 8 & Fig. 9. Data analyses is based on three biological replicates. Each Biological replicate had three internal replicates.

Click here for additional data file.

Supplemental Information 8 The NGSS Life Science disciplinary core ideas align with the 12 principles of Plant Biology from ASPB

Click here for additional data file.

Supplemental Information 9 Article published from our lab in the NCUR Proceedings centered on Chlamydomonas 10E35/lsr1a mutant

Click here for additional data file.

Table S1 pH changes in dark-and light-exposed vials containing Chlamydomonas 4A+ strain beads in de-ionized (DI) water and tap water

The table shows the mean pH with standard deviations based on data from three biological replicates. Phenol red was used as the pH indicator in the control and experimental vials. pH was measured using a pH meter. Raw pH data of three biological replicates with statistical analyses can be found in https://doi.org/10.6084/m9.figshare.12344024.v1 and in the Data S1 file. Data S2 file contains the raw pH data with mean and standard deviation information. Each biological replicate had three internal replicates.

Click here for additional data file.

Table S2 pH changes in dark-and light-exposed vials containing Chlamydomonas 4A+ strain beads differing in cell numbers per bead in tap water

The table shows the mean pH with standard deviations based on data from three biological replicates. Phenol red was used as the pH indicator in control and experimental vials. Cell numbers shown below are approximate estimates. pH was measured using a pH meter. Raw pH data of three biological replicates with statistical analyses can be found in https://doi.org/10.6084/m9.figshare.12344024.v1 and in the Data S1 file. Data S2 file contains the raw pH data with mean and standard deviation information. Each biological replicate had three internal replicates.

Click here for additional data file.

Table S3 pH changes in the 4A+ bead bracelet that was exposed to light/darkness for three hours

The table shows the mean pH with standard deviations based on data from three biological replicates. Control algal bracelet was not exposed to light or to darkness (zero time point). Bicarbonate indicator was used as the pH indicator in the algal bead bracelets. pH was measured using pH testing strips. Raw pH data of three biological replicates with statistical analyses can be found in https://doi.org/10.6084/m9.figshare.12344024.v1 and in the Data S1 file. Data S2 file contains the raw pH data with mean and standard deviation information. Each biological replicate had three internal replicates.

Click here for additional data file.

Table S4 pH changes in 9 hours- and 15 hours-dark adapted bracelets that were shifted to light for 4 hours and 12 hours, respectively

The table shows the mean pH with standard deviations based on data from three biological replicates. Bicarbonate indicator was used as the pH indicator in the algal bead bracelets. pH was measured using pH testing strips. Raw pH data of three biological replicates with statistical analyses can be found in https://doi.org/10.6084/m9.figshare.12344024.v1 and in the Data S1 file. Data S2 file contains the raw pH data with mean and standard deviation information. Each biological replicate had three internal replicates.

Click here for additional data file.

Table S5 pH changes in dark-and light exposed-vials containing wild type 4A+ and 10E35 mutant beads in tap water

The table shows the mean pH with standard deviations based on data from three biological replicates. Phenol red was used as the pH indicator in control and experimental vials. pH was measured using a pH meter. Raw pH data of three biological replicates with statistical analyses can be found in https://doi.org/10.6084/m9.figshare.12344024.v1 and in the Data S1 file. Data S2 file contains the raw pH data with mean and standard deviation information. Each biological replicate had three internal replicates.

Click here for additional data file.

Table S6 Cellular respiration-induced pH changes in the dark in 4A+ and 10E35 bead vials that were exposed to light for 4 hours

The table shows the mean pH with standard deviations based on data from three biological replicates. Phenol red was used as the pH indicator in control and experimental vials. pH was measured using a pH meter. Raw pH data of three biological replicates with statistical analyses can be found in https://doi.org/10.6084/m9.figshare.12344024.v1 and in the Data S1 file. Data S2 file contains the raw pH data with mean and standard deviation information. Each biological replicate had three internal replicates.

Click here for additional data file.

Table S7 Photosynthesis-induced pH changes in the light in 4A+ and 10E35 bead vials that were exposed to dark for 6 hours

The table shows the mean pH with standard deviations based on data from three biological replicates. Phenol red was used as the pH indicator in control and experimental vials. pH was measured using a pH meter. Raw pH data of three biological replicates with statistical analyses can be found in https://doi.org/10.6084/m9.figshare.12344024.v1 and in the Data S1 file. Data S2 file contains the raw pH data with mean and standard deviation information. Each biological replicate had three internal replicates.

Click here for additional data file.

The authors would like to thank Dr. Krishna K. Niyogi (Professor, Department of Plant and Microbial Biology, UC Berkeley, CA) for providing us with the Chlamydomonas wild type strain 4A+. We would also like to thank the following science teachers and instructors who allowed us to implement our lab activities in their classrooms at their institutions: Dagmah Alexander and Dione Belser (Camp Creek Middle School, GA), Gini Loeffler (East Coweta Middle School, GA), Tim Hawig (Carrollton High School, GA), Chrissy Loveless (Carrollton Junior High School, GA), Darrius Shaw (Crabapple Middle School, GA), Patricia Thrower (Bremen Middle School, GA), Kallendra Mathews (Bremen High School, GA), Keisha Barnes (Lithia Springs High School, GA), Gina Watkiss (The Heritage School, GA), Emily Camp (The Heritage School, GA), Karin van den Hoonaard (The Heritage School, GA), Ryan Danbury (The Heritage School, GA) and Phillip Grovenstein (Lab Coordinator and Instructor, Perimeter College, Life & Earth Sciences, Georgia State University, GA). We are grateful to Mr. Taylor Box (Biology major, University of West Georgia, GA), Miss Victoria Johnson (Elementary school student, Jones Elementary School, GA) and Dr. Melissa Johnson (Associate Professor in Biology, University of West Georgia, GA and mother of Miss Victoria Johnson) for their participation in the Chlamydomonas bracelet making demonstration videos.

Additional Information and Declarations

Competing Interests

Author Contributions

Data Availability

The authors declare there are no competing interests.

Mautusi Mitra and Sara Michelle Broom conceived and designed the experiments, performed the experiments, analyzed the data, prepared figures and/or tables, authored or reviewed drafts of the paper, and approved the final draft.

Kysis Pinto conceived and designed the experiments, performed the experiments, analyzed the data, prepared figures and/or tables, and approved the final draft.

Sovi-Mya Doan Wellons conceived and designed the experiments, performed the experiments, prepared figures and/or tables, and approved the final draft.

Ariel Dominique Roberts performed the experiments, analyzed the data, authored or reviewed drafts of the paper, and approved the final draft.

The following information was supplied regarding data availability:

Raw pH data are available as Supplemental Files and at Figshare: Mitra M, Broom S, Pinto K, Wellons S-MD, Roberts AD. 2020. pH data with statistical analyses from experiments under light and darkness using Chlamydomonas reinhardtii bead bracelets and glass vials containing Chlamydomonas beads. DOI: https://doi.org/10.6084/m9.figshare.12344024.v1.

The research protocol for making algae bead bracelets/necklaces are at protocols.io: Mautusi Mitra, Sara Broom, Kysis Pinto, Sovi-Mya Doan Wellons, Ariel Dominique Roberts 2020. Making inexpensive light-powered Chlamydomonas reinhardtii (a green micro-alga) bead bracelets/necklaces for teaching the interplay of photosynthesis and cellular respiration to K4-K16 students. protocols.io. DOI: https://doi.org/10.17504/protocols/io/bgpyjvpw.

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
