# Peer review of "Engaging inexpensive hands-on activities using Chlamydomonas reinhardtii (a green micro-alga) beads to teach the interplay of photosynthesis and cellular respiration to K4–K16 Biology students"

_PeerJ, doi:10.7717/peerj.9817_

## Round 0.1 · original submission · Major Revisions

Please take into consideration the reviewer's comments, and provide a revised manuscript and a detailed point-by-point rebuttal letter.

Reviewer 1 ·

Basic reporting

I find the manuscript mostly clear and unambiguous. I note some areas that could use clarity in that they may be missing a word, need greater or less explanation. It may be a bit more editorial than necessary. Additionally, I list a few questions regarding the introduction below. References and background seem sufficient, unless noted below.

Questions regarding the introduction:
Could the authors explain the advantage of using a motile strain of green algae vs. non motile? I’d like to know if it is the novelty of being able to expand the species of unicellular algae that can be studied, using a well-known model organism, etc. – I see that this is in the discussion, but I think it is helpful to have a brief reference in the introduction as well.
Is there evidence for the available kits not working well? (ie. did your group test them? Survey K16 teachers?)

Writing clarity:
Lines 23-25: awkward wording, suggestions in red: These kits are expensive, can be applied to only non-motile algae, and provide protocols that lack detailed specifics for trouble shooting and does not provide customization guidelines for different grade levels.
Lines 47-49: awkward wording, suggestions in red: Our comparative studies of photosynthesis and cellular respiration in the10E35 and in 4A+ strains showed that 10E35 relatively has a higher respiration rate and a lower photosynthetic rate than 4A+.
Line 90: maybe add “fun and engaging way” to sound more technical?
Line 93: “conduct their own independent experiments” – this shows the activity is not just a ‘cookie cutter’ lab.
Line 95: remove “in” before “in dark”
Lines 98-100: It seems redundant to have both “functionally uncharacterized” and “unknown function” in this sentence.
Lines 105-107: Add a statement that acknowledges that cellular respiration is still going on when the cells are in the light, but the net production of O2/use of CO2 from photosynthesis vastly outweighs the O2 consumption/CO2 release during cellular respiration (a very common misconception is that cellular respiration doesn’t occur in the light)
Lines 113-116: this is just a very long sentence. Maybe according to grade level, availability of resources, and allow better time management?

Experimental design

I appreciate the level of detail that went into validating methods for this new teaching tool. I have a few questions below and some suggestions for clarity for the materials/methods provided for those using this for their own teaching.

For clarification – when referring to biological replicate – you are meaning 3 separate replications of the experiment, each with 3 internal replicates? Apologies, I got lost in the large amount of data presented in the supplemental data files. If this is not the case, I would suggest having more replicates to strengthen your findings.


dx.doi.org/10.17504/protocols.io.bgpyjvpw/ Are there copyright issues with having the protocol previously published?

Text_S1
Item 21: aluminum foil
Item 30: it seems that calling it a hemocytometer or counting chamber might be more flexible to users?

Text_S2
In order to have the greatest possible flexibility to educators likely working on a tight budget or who may already have some of these supplies, it would be helpful to have a disclaimer with the supply list that there may be other vendors for the general supplies that are cheaper or that they already work with.
The easiest way to do this is adding a column to the left that has the generic term for the item (ie. 1.5 ml centrifuge tubes, yarn, disposable inoculating loops and needles, hemocytometer, etc.) as they may already have these items/be able to find cheaper. For example, the Tygon E3603 laboratory tubing is $211 (I realize you get a ton of tubing, but maybe a school would only want a small amount and look to pick it up at a hardware store) and it could be broadly represented by providing the dimensions.

It may be beneficial to provide a basic protocol on how to use a hemocytometer in a teaching setting? Again, this could be a supplemental text, similar to S1 and S2.

Validity of the findings

Analysis of data appears robust (provided the replicates are as stated in section 2) and statistically sound. A few specific comments:

Lines 280-282: I don’t typically see p values referred to as percentages in scientific papers? Additionally, some of this information regarding the type of tailed test could be removed to condense the results and included in the figure legends (and kept in the data analysis section of the methods) to condense the text in the results.

Thinking of this article in terms of describing teaching activities, it would be helpful to have a figure/supplemental text for specifically trouble shooting common problems.

I think in terms of audience that teachers using this activity would benefit from the separation of technical details from the learning activities, pre-class assignments/background, assessments aligned to learning goals. Perhaps this could be combined with trouble shooting and supplies into a single supplemental text that is information on teaching?

Some of the items in the discussion, such as customization by grade level, could be in their own figure to highlight and provide more options (as it sounds like you have them with 14 other activities!) - this could be similar to Table 9.

Additional comments

This manuscript describes a holistic picture – using basic scientific research to create engaging activities for K4-16 students. I very much appreciate that this work was generated in collaboration with high school and undergraduate students.

This manuscript has a huge amount of data! For me, there is important information that has two broad audiences and you will not be able to reach them in a single manuscript. Review instructions state that decisions are not made based on any subjective determination of niche audience, so this may be out of line, but the work seems like 2 different manuscripts that differ on scope and audience: 1) a methods paper providing all data (potentially include data from verification of the other developed activities?) with optimizing this measure of photosynthesis and cellular respiration - color change, pH, time course experiments, and statistical analysis – geared towards the broader scientific community and fitting in the PeerJ journal, and 2) a teaching module designed for K4-16 educators that has the protocols, and much less technical information (suggestions include International Journal of STEM Education, Frontiers in Education: STEM Education, American Biology Teacher).

If I am incorrect with regards to my assessment of the scope of PeerJ, I would advise a greater separation between teaching-specific information (simplicity and clarity) and basic data. If PeerJ looks for this type of comprehensive work, I applaud having a platform that is open access and can appeal to multiple, historically separated, audiences.

·

Basic reporting

Here, the authors designed a useful and inexpensive protocol as a tool to teach in a fun and easy way the basic concepts of photosynthesis. Work is of great importance at the educational/academic level. However, some points can be improved, such as the ones I point out below:
-I suggest deepening in the basic concepts of the photosynthesis process and highlight its biological importance. This could help you to strengthen your work justification.
-I suggest a change in the order of information in the introduction section. Line 87 to 101 is about your results, so, I think that this information should appear after line 110.
-There is information that doesn't have references, please include the original works from which that information was taken. i.e., lines 103, 107, 334, 541.
-The link that you provide for “Carolina QuickTips Making Algae Beads” reference doesn't work, please check it.
-It's highly recommended that the figures show all the necessary data to can understand them even without seeing the figure legend. In this regard, I suggest including some missed information in the figures. For example, in Fig. 1, please indicates which vials correspond to DI or tap water; Fig 2, please indicates the concentrations of the beads, etc. For more details please check the attached pdf file.
-Tables 1 to 7 are repeated or complementary information of figures, thus, I suggest sending them to supplementary material.

Experimental design

The information obtained in this investigation has very educational and academic significance, which will be helpful for many educators and institutions.
The research question was well defined and I think that the methods used were appropriate. Also, these were described in detail. I only have a few suggestions:
-Considering the nature of your investigation and intending to facilitate the reading of your results, I suggest adding one figure to show the color-scale expected when you use the pH indicators as the bicarbonate indicator and the phenol red solution.
-I suggest describing that phenol red solution is used as a pH indicator too, as a bicarbonate indicator (line 236).
-For the evaluation of the effect of cell density in line 234, I suggest writing the two concentrations of beads used.

About the data analysis, I appreciate that you provide all the statistical analysis, however, I found them difficult to read. I suggest founding a better way to describe them. For instance, you can consider the way you describe it on lines 378-379. In this example, you summarize the information and put them in parentheses. Also, consider that it's not mandatory to describe all the numeric data obtained in your statistical analysis. Because you provide all your analysis in detail in supplementary material; you can always refer to that.

Validity of the findings

I consider that all the experiments here described were well performed, and the results are supported by statistical analysis. The conclusions are well stated.

Additional comments

Please refer to the attached file for more details.

Reviewer 3 ·

Basic reporting

Overall, the paper has a logical flow of ideas and is self-contained. There are some grammatical issues in the paper, in particular those around use of tense. The Background section of the Abstract would benefit from an additional sentence connecting the ideas from sentence1 to 2 (lines 20-23). The third sentence in the Abstract Background has grammatical problems and would benefit from the use of colons and semicolons (lines 23-25). It is unclear to the rationale for including the last sentence of the Abstract Background (lines 28-31). The article is missing in the second sentence of the Methods portion of the Abstract (lines 24-36). There are grammatical issues with the last sentence of the Abstract Methods (lines 36-39). Tense and verb-subject agreement issues in the second sentence of the Results section of the Abstract (lines 41-44). The fourth and last sentences of the Abstract Results section are a different tense from the rest of the section (lines 45-47, 50-52). The first sentence of the Introduction needs commas separating the coordinating adjectives and the sentence ends with a phrase that is not grammatically correct (lines 55-56). The second sentence of the Introduction is a run on and would benefit from being split into two (lines 56-60). These are some of the initial edits that should be made.

In the Introduction section, BIO-RADs “Photosynthesis and Cellular Respiration Kit for General Biology” should be added to the list of currently available commercial kits, as it offers the most comprehensive activity guidelines and includes detailed troubleshooting prompts (lines 60-64, 70-72). It is unclear what the advantage would be of using motile versus non-motile algae for photosynthesis and respiration assays (lines 67-70). Authors should articulate the rationale for use of Chlamydomonas over Chlorella, as the commercial kits use the latter. Related, the authors should explain the advantage of making the beads over purchasing pre-made beads, such as the ones available from BIO-RAD. Given the number of additional reagents and time needed for making the beads, it is unclear if this protocol is more cost effective in comparison to the BIO-RAD kit, which one can purchase for $181 in the US. The author’s should include a detailed cost comparison to support this claim. The authors should read Canbay, Kose, and Oncel (2018) for a published protocol for making algae beads using Chlamydomonas (lines 67-68).

The figures are extensive and well organized. For the sake of reducing confusion in your target audience (K4-12 teachers), it is recommended that the hand-written labels on the glass vials be cropped out. Given the importance of subtle color changes, the images should have a consistent white-balance applied. The authors are encouraged to look at Figure 2(A), specifically.

Experimental design

It is unclear if the focus of this article is the modification of existing protocols for use with Chlamydomonas or the use of this technique as part of a teaching module. If the former, then I would encourage the author’s to read the paper by Canbay, Kose, and Oncel (2018) that describes production and analysis of immobilizing Chlamydomonas in beads. The author’s should describe how their protocol differs and/or is superior to the protocol described in the 2018 paper. If the latter, then it is unclear why the author’s would choose to include a mutant strain that is not available from the Chlamy Stock Center. The authors do include a list of strains that are available, but results from this activity using those strains was not provided (lines 544-547). For a K6-12 teacher, it will be difficult for them to know what expected results are. The authors can provide a table or figure describing expected outcomes using exemplar strains available from the Chlamy Stock Center.

In the “Preparation of Chlamydmomonas…” subsection, the URLs do not work because of the inclusion of the forward slash at the end of the link (lines 160 and 166). The chemical analysis of the “tap water” should be included to define “tap water” for this study and confirm that chemicals that would modify photosynthesis/respiration, such as Acetate or Sulfur-containing compounds, are not present.

Validity of the findings

This paper modifies a published protocol and applies it to existing activities with a new twist, which is the plastic tubing “bracelets”. The tubing allows for better visualization of the pH changes due to the activity of organisms in individual beads, in comparison to glass vials or microcentrifuge tubes. The “bracelet” modification is meant to capture the attention of students, and this should be effective in doing so, however, data was not provided. The existing, commercially-available beads can also be repackaged into this “bracelet” form. The authors sufficiently addressed the caveats of the plastic tubing.
Given the number of institutions this protocol has been used at, the inclusion of student and teacher opinions would have strengthened the argument that the learning module is engaging. The authors also claim that this protocol is most cost-effectives than existing protocols or kits. The authors should include a detailed list of all reagents and equipment required, with their respective cost of use, to support this claim.

---

## Round 0.2 · Minor Revisions

Please take into consideration the reviewer’s comments and provide back a point-by-point rebuttal letter addressing those concerns.

·

Basic reporting

The authors followed the recommendations made to improve their work.
The introduction became more complete and clear with the changes made. Also, the figures were improved and now is easier to see what they want to show.
I appreciate that you move the tables to supplemental material. In this way, it is easier to read the manuscript.

Experimental design

The changes made to the descriptions of statistical analyses improve considerably your manuscript. However, I think that it is very important to mention which was the p-value used in your analyses. You can mention it in the "Imaging and Data analyses" section or, you can point it out in the results section, by enclosing the p-value in parentheses when necessary. Nevertheless, it is important to note that the p-values usually are presented as decimals, as already mentioned by another reviewer. So, I suggest writing in these ways in the manuscript, and if you want to conserve the percentages data in the Data files, you can do it, too.

Validity of the findings

I consider that this work will have an important impact on the educational and academic level.
I think that the experiments were properly developed and the conclusions are well stated. Also, the changes made to the "Materials and methods" section improve considerably the manuscript.

Additional comments

I appreciate the changes made. I think you improved the manuscript a lot. I consider that it is almost ready for publication.

---

## Round 0.3 · accepted · Accept

Thanks for addressing the minor revisions requested. Now your manuscript is accepted in PeerJ.